# Parametric portfolio policy with momentum-based sentiment trading strategy

**Wen-Yi Lee[1], Yu-Hsuan Lin[1], Jing-Rung Yu[ID][2]\*, Donald Lien[ID][3]**

**1** Department of Information Management, National Taipei University of Business, Taipei, Taiwan,
**2** Department of Information Management, National Chi Nan University, Nantou, Taiwan, **3** Department of Statistics and Data Science, University of Texas at San Antonio, United States of America

\* jennifer@ncnu.edu.tw

## Abstract

To enhance the effectiveness of the conventional mean-variance portfolio model, this study introduces a parametric portfolio policy that incorporates a momentum-based sentiment characteristic vector. This vector enables the identification of outperforming assets by capturing both historical returns and market sentiment. Drawing on a decade of rebalancing data from the S&P 500 and Dow Jones 30 constituent stocks, the proposed model optimizes the interrelationships among portfolio holdings, a benchmark portfolio, and the constructed characteristic vectors. In contrast to conventional static back testing approaches, the proposed model accounts for transaction costs and is evaluated over a 15-year investment horizon. Empirical results demonstrate that the proposed model significantly outperforms the benchmark, particularly the minimum-variance model that does not incorporate sentiment-driven parametric adjustments. During periods of financial crisis, the model selects sentiment-based momentum more frequently, leading to differing asset allocations and potentially higher utility for investors. The sentiment-augmented momentum strategy exhibits superior performance compared to the conventional mean-variance approach. The findings underscore the importance of integrating market sentiment into characteristic vector construction, affirming the value of parametric portfolio policies in improving asset allocation and risk-adjusted returns.

## 1. Introduction

Portfolio optimization is crucial for allocating resources to assets. The mean-variance portfolio (MV), proposed by Markowitz [1], minimizes portfolio variance at a given required return. To enhance the performance of the MV, Brandt et al. [2] developed characteristic vectors using structured data such as historical returns and book-to-market ratios. The characteristic vector with equal weights for high-return stocks is derived from momentum strategy principles. The characteristic vector assets can be selected based on the momentum effect [3], surplus consumption ratio [4], or

**Data availability statement:** All relevant data are within the paper and Supporting Information files.

**Funding:** Funding recipient: Wen-Yi Lee Grant number: NSTC 111-2410-H-141-004-The National Science and Technology Council, Taiwan URL: https://www.nstc.gov.tw/ The sponsors or funders did not play any role in the study design, data collection and analysis, decision to publish, or preparation of the manuscript.

**Competing interests:** The authors have declared that no competing interests exist.

technical indicators [5]. Recently, Ko et al. [6] proposed the Fama–French three-factor model as a characteristic vector to enhance portfolio performance, indicating that this vector can enhance portfolio performance.

Brandt et al. [2] suggest that portfolio construction using the parametric portfolio policy (PPP) involves a trade-off of three dimensions: the holding portfolio, benchmark portfolio, and characteristic vectors. Jiang et al. [7] applied the equally weighted strategy to integrate the characteristic vector into the MV portfolio. Chen and Fei [8] found that momentum, firms' book-to-market ratios, and size are crucial when constructing characteristic vectors in the Chinese market. Furthermore, time-varying characteristic vectors are vital for improving portfolio performance. They suggested that portfolios incorporating time-varying PPP in asset allocation outperform those that do not. Caldeira et al. [9] extended PPP to address the non-linear relationship between firms' returns and characteristics.

Recent advancements in portfolio research highlight the versatility of parametric portfolio policies in improving asset allocation outcomes. Zeng et al. [10] found that generalized linear models enhance momentum strategies by integrating firm characteristics to capture return persistence more effectively. Golosnoy et al. [11] demonstrated that combining competing prediction rules within a parametric structure enhances the stability of global minimum variance portfolios, particularly by detecting risky assets. Beyond linear forms, Li et al. [12] proposed nonlinear neural network–based policies that directly map firm characteristics into portfolio weights, while applying heuristic smoothing to control turnover and transaction costs. Arasteh [13] added a multi-period dimension by incorporating real options and ARIMA–GARCH forecasts, highlighting the adaptability of parametric policies under uncertainty. These studies demonstrate that parametric portfolio policies, whether linear, nonlinear, or dynamic, provide a coherent framework for improving predictive accuracy and economic feasibility in portfolio optimization.

However, such studies have constructed characteristic vectors exclusively using structured data, such as numerical indicators from historical returns, without incorporating market sentiment into their trading strategies. Recent studies have explored the use of unstructured data, such as news articles, for sentiment analysis in the development of trading strategies [14–16]. Specifically, news articles influence investors' perspectives, a phenomenon known as market sentiment [17–21]. Since the sentiment expressed in news articles influences stock market trends, sentiment analysis can improve profitability derived through the momentum strategy [22]. However, to our knowledge, the application of sentiment analysis in constructing characteristic vectors is limited. Most studies focus on building characteristic vectors with structured data [2,23–25] rather than unstructured data. Thus, our study developed a momentum-based sentiment characteristic vector that selects winners based on asset returns, followed by sentiment analysis to identify value-added investment assets.

Nevertheless, current research does not account for trade costs [2,3,8,9] while incorporating the PPP into a portfolio model. However, this would simplify the process by treating each model as static rather than considering the friction cost associated with actual transactions. As such, this study rebalances portfolios with trade cost

minimization while optimizing the relationship between the target portfolio and momentum-based sentiment characteristic vector. The proposed model generates a compromised trade-off of asset allocation among the holding portfolio, benchmark portfolio, and the momentum-based sentiment characteristic vector.

This study offers three advantages. First, it develops a momentum-based sentiment characteristic vector that transforms the textual information of assets and sequentially ranks their historical returns. Second, it optimizes the weighting trade-off of the portfolio among the asset holding, the benchmark portfolio, and the characteristic vector. Unlike the static model of Behr et al. [3], our proposed portfolio model rebalances based on the prior period's holding weight and characteristic vector [2,3]. Finally, the proposed model can diversify portfolios through characteristic vectors, which addresses the conventional MV model's tendency to concentrate investments on fewer assets.

The results indicate that the proposed portfolio model with momentum-based sentiment effect (MV_MS) outperformed the portfolio model based upon mean-variance with momentum (MV_M) and MV using the S&P 500 and Dow Jones 30 datasets. The MV_MS achieved the highest Sharpe ratio compared to the MV, MV_M, and the market. This finding implies that incorporating momentum-based sentiment characteristic vectors enhances the performance of the MV model beyond that achieved by considering return momentum alone. Since the MV_MS and MV_M outperform the MV model and the market, we examined both models using data from the financial crisis period. The results demonstrate that the portfolio with a momentum-based sentiment vector performed more effectively than the MV_M model with the S&P 500 and Dow Jones 30 datasets; the MV_M model experienced significant losses in final market value with the Dow Jones 30 dataset.

The remainder of this paper is organized as follows. Section 2 introduces the parametric portfolio policy. Section 3 details the proposed mean-variance model with the momentum-based sentiment characteristic vector (MV_MS). Section 4 presents the experiment and comparison. Section 5 summarizes the study conclusions.

## 2. The parametric portfolio policy

The parametric portfolio policy (PPP) proposed by Brandt et al. [2] considers the trade-off between the benchmark portfolio and characteristic vector. Behr et al. [3] modified PPP using a utility function to optimize the importance between the benchmark portfolio and the characteristic vector. Let us assume there are $N$ assets. Let $\hat{m}_i$ be the weight of the $i$-th asset chosen according to the momentum or momentum-based sentiment effects; this represents the characteristic vector, also called the winner portfolio. Under a no-short-sale condition, the values of the characteristic vector are $1/N_w$ for the assets within the winner portfolio and 0 for the other assets, as shown in Eq. (1).

$$\hat{m}_i = \begin{cases} \frac{1}{N_w}, & \text{if the } i\text{–th asset is in the winner portfolio} \\ 0, & \text{otherwise} \end{cases},$$

(1)

where $N_w$ represents the number of assets in the winner portfolio, ranked according to each asset's average return for the momentum strategy. Since the momentum strategy follows the equal weight policy for the winner portfolio [26], $\hat{m}_i$ is equal to $1/N_w$ for the assets within the winner portfolio. $1/N_w$ indicates that all winners have the same influence on the benchmark portfolio. The sum of characteristic values equals 1, indicating that the characteristic vector has a stationary distribution over time. In the momentum strategy, the winner portfolio comprises assets with the top 10% cumulative historical returns. The value of each asset is equal to $1/N_w$. To determine the selected assets, cumulative historical returns are used to identify the top decile. The winner portfolio composition varies between the momentum and momentum-based sentiment effects. In the proposed sentiment-based momentum strategy, the winner portfolio comprises assets within the top 30% with the highest returns, and selects the top 10% based on positive sentiment among the top 30% highest-returning assets.

Eq. (2) generates the weight vector of the target portfolio ($w^{target}$) by integrating the benchmark portfolio ($w^b$) and characteristic vector ($\hat{m}$). $w^b$ is obtained by optimizing the target portfolio, such as the MV. The components of $\hat{m}$ follow

Eq. (1), assigning $1/N_w$ to the assets within the winner portfolio. The sizes of the weight and characteristic vectors are $n$ x 1, where $n$ indicates the number of investment assets. Based on the PPP, $\theta$ is the decision variable trade off between the benchmark portfolio and characteristic vector (Eq. 2). $\bar{r}$ indicates the vector of average return. After obtaining $\boldsymbol{w}^b$ and $\hat{\boldsymbol{m}}$ by implementing the benchmark portfolio and momentum strategy, Eq. (3) is used to maximize the utility function, referencing Behr et al. [3], which results in $\boldsymbol{w}^{target}$.

$$w^{target} = (1-\theta)w^b + \theta\hat{m}, \tag{2}$$

$$\max_{\theta} u\left(\left(w^{target}\right)^{\mathsf{T}}\bar{r}\right) = u\left(\left((1-\theta)w^b + \theta\hat{m}\right)^{\mathsf{T}}\bar{r}\right). \tag{3}$$

While incorporating the utility function in Eq. (3), a quadratic utility function is considered, as shown in Eq. (4) following Behr et al. (2012).

$$\max_{\theta} (1-\theta)r_b + \theta r_m - \frac{\gamma}{2}((1-\theta)r_b + \theta r_m)^2 \tag{4}$$

s.t.

$$r_b = \left(w^b\right)^{\mathsf{T}}\bar{r}, \tag{5}$$

$$r_m = \left(\hat{m}\right)^{\mathsf{T}}\bar{r}, \tag{6}$$

where $r_b$ is the benchmark portfolio return and $r_m$ is the return in the characteristic vector. $\theta$ is determined by maximizing the utility function in Eq. (4). According to Brandt et al. [2], the linear combination of target and holding portfolios represents the portfolio weights ($\boldsymbol{w}$), as shown in Eq. (7), which is used for statistical comparison. The holding portfolio weight is calculated from the previous investments and is, therefore, regarded as the weight prior to rebalancing.

$$w = \alpha w^h + (1-\alpha)w^{target}, \tag{7}$$

where $\alpha$ is a constant, assumed to be 0.5, which makes a trade-off between the holding and target portfolios in this static model. $\boldsymbol{w}$ represents the rebalanced portfolio weight vector, which is derived by integrating the holding and target weight vectors. While Behr et al. [3] made comparisons based on the static portfolio model, we revise Eq. (7) to enable dynamic rebalancing between the holding and target portfolios rather than only adjusting $\alpha$ in a static portfolio model.

## 3. The proposed mean-variance with momentum-based sentiment characteristic vector (MV_MS)

When the momentum effect is considered, as per Behr et al. [3], the winner set comprises the top 10% of stocks ranked by their accumulated returns over the previous six months [26]. However, when the momentum-based sentiment effect is considered, the top 30% winners are initially selected based on their accumulated returns. Then, the top 10% winners are chosen from this subset based on the sentiment score [22]. Since the experiments are performed based on monthly rebalancing, the holding weights ($\boldsymbol{w}^h$) are derived from the previous period, and the benchmark weights ($\boldsymbol{w}^b$) are generated from the benchmark model, the mean-variance model. $\hat{\boldsymbol{m}}$ is derived from either the momentum or the momentum-based sentiment strategies.

Thus, differing $\hat{m}$ values are obtained by applying various trading strategies. This approach provides a straightforward and adaptable method for integrating various trading strategies. The MV_M utilizes the $\hat{m}$ derived from the momentum strategy, while the proposed MV_MS employs the $\hat{m}$ obtained from the momentum-based sentiment trading strategy. Both MV_M and MV_MS aim to minimize the trade costs through rebalancing. The proposed model MV_MS, without short-selling, is as follows:

$$\underset{\theta,\, \boldsymbol{l}^+,\, \boldsymbol{\Gamma}}{\text{Min}}\ p_1 \mathbf{1}^{\mathsf{T}} \boldsymbol{l}^+ + p_2 \mathbf{1}^{\mathsf{T}} \boldsymbol{\Gamma} - (1-\theta) r_b - \theta r_m + \frac{\gamma}{2}\left((1-\theta) r_b + \theta r_m\right)^2 \tag{8}$$

s.t.

$$\mathbf{1}^{\mathsf{T}} \boldsymbol{w} + p_1 \mathbf{1}^{\mathsf{T}} \boldsymbol{l}^+ + p_2 \mathbf{1}^{\mathsf{T}} \boldsymbol{\Gamma} = 1, \tag{9}$$

$$\boldsymbol{w} = \boldsymbol{w}^h + \left((1-\theta)\boldsymbol{w}^b + \theta\hat{\boldsymbol{m}}\right) + \boldsymbol{l}^+ - \boldsymbol{\Gamma}, \tag{10}$$

$$\boldsymbol{w}^{\mathsf{T}} \bar{\boldsymbol{r}} \geq c, \tag{11}$$

$$0.001\boldsymbol{u} \leq \boldsymbol{w} \leq \boldsymbol{u}, \tag{12}$$

$$\boldsymbol{u} \in \{0, 1\}, \tag{13}$$

$$0 \leq \theta \leq 1, \tag{14}$$

$$\mathbf{0} \leq \boldsymbol{l}^+, \boldsymbol{\Gamma} \leq \mathbf{1}, \tag{15}$$

Constraints (5) and (6),
where $p_1$ and $p_2$ indicate the transaction fee proportions for buying and selling assets, assumed to be 0.25% for rebalancing. $\theta$ is the decision variable that trades off the benchmark portfolio and characteristic vector based on the PPP. $\gamma$ is assumed to be 30, following Behr et al. (2012). Eq. (8) is the objective function, minimizing trade costs while maximizing utility, with $\theta$, $\boldsymbol{l}^+$, and $\boldsymbol{\Gamma}$ as decision variables. Eq. (9) indicates that the allocated asset weights and their corresponding transaction cost proportions are equal to 1. Eq. (10) encompasses the holding weight $\boldsymbol{w}^h$ and $\boldsymbol{w}^{target}$ from the benchmark weights ($\boldsymbol{w}^b$) and the characteristic vector weights ($\hat{\boldsymbol{m}}$). Compared to the static model in Brandt et al. (2009), $\boldsymbol{l}^+$ and $\boldsymbol{\Gamma}$ in Eq. (10) indicate the dynamic buying and selling propositions for each rebalancing.

Instead of using the static model from Section 2, $\alpha$ is omitted and $\boldsymbol{w}$ is generated dynamically according to the proposed rebalancing model. In Eq. (11), $c$ denotes a predetermined threshold of the required return for the rebalanced portfolio. The rebalanced portfolio required return ($c$) is set as 0.1% in this study. Eq. (12) sets the upper and lower bounds for the weight of the invested asset.

Unlike Behr et al. [3], who focuses on the influence of $\boldsymbol{w}^{target}$, this model extends a rebalancing model. The objective function considers minimizing trade costs and maximizing the portfolio return of trading strategy and utility.

## 4. Experimental results

To examine the proposed model's performance, this study utilized S&P 500 and Dow Jones 30 composite stocks from January 1, 2005 to January 4, 2022. After omitting stocks with missing data, 382 stocks from the S&P 500 and 27 stocks from the Dow Jones 30 are included in the analysis. The estimated expected returns ($\bar{r}$) are calculated as the average of 60 daily historical returns, with a holding period of 20 transaction days for each rebalance. The first portfolio rebalancing occurred on January 26, 2005, and ended on January 4, 2022. A rolling window of 20 transaction days was employed, resulting in 220 rebalances. $\alpha$ is specified as 0.5 (S1 Appendix).

To collect unstructured data, the Wall Street Journal is selected due to its prominence as a financial newspaper among American investors. News articles can significantly impact a company's stock price, as indicated by Tetlock [17], Li et al. [27], Hao et al. [28], Hsu et al. [29], Usmani and Shamsi [30], and Hung et al. [31]. News articles from the Wall Street Journal published between January 1, 2005 and November 17, 2022, were gathered; daily sentiment scores were obtained for each composite stock.

The sentiment score for each article related to corresponding stocks was calculated using the Python package NLTK [32–34]. The average news sentiments within a day represent the market sentiment of the corresponding company on that day. Following Kim and Suh [22], the top 30% of assets ranked by the accumulated return over 20 trading days were selected. Aiming at assets with high return and high sentiment, the top 10% of these selected assets were ranked as winners based on the average sentiment score within six months. Thus, in the sentiment characteristic vector, winners are denoted as "$1/N_w$," indicating buy-in, while other assets set are as "0," indicating no investment.

Table 1 compares the rebalancing models: the proposed MV_MS against MV_M and MV. The S&P 500 and Dow Jones 30 index returns serve as additional benchmarks. The MV model, buy and hold, and naïve strategies serve as alternative benchmark models. Initially, six months (120 transaction days) are used to form the initial portfolio [22]. Portfolios are rebalanced every 20 trading days.

In Panel A, using the S&P 500 dataset, the proposed MV_MS exhibits the highest Sharpe ratio compared to MV_M and MV. The results demonstrate that MV_MS enhances the market value of MV_M, even after considering the trade costs. Specifically, MV_MS achieves a return 0.24% higher than MV_S and 0.35% higher than MV. The Sharpe ratio of

**Table 1. Portfolio comparison using the S&P 500 and the Dow Jones 30 datasets (January 1, 2005 to January 4, 2022).**

| | Realized | | | Market value | |
|---|---|---|---|---|---|
| | Average Return (%) | Standard deviation (%) | Sharpe ratio | Final | Average |
| Panel A: S&P 500 | | | | | |
| MV | 0.94% | 5.30% | 0.177 | 5,145,086 | 2,100,427 |
| MV_M | 1.05% | 5.18% | 0.203 | 6,597,270 | 2,696,578 |
| MV_MS | 1.29% | 5.19% | 0.249 | 10,788,043 | 3,713,499 |
| Buy and hold | 1.06% | 4.94% | 0.215 | 7,686,905 | 3,427,768 |
| Naïve | 1.02% | 5.05% | 0.203 | 6,988,687 | 3,123,605 |
| S&P 500 index (S&P500) | 0.04% | 1.23% | 0.033 | | |
| Panel B: Dow Jones 30 | | | | | |
| MV | 0.95% | 4.09% | 0.232 | 6,003,082 | 2,657,274 |
| MV_M | 1.13% | 4.23% | 0.267 | 8,575,916 | 3,511,711 |
| MV_MS | 1.24% | 4.57% | 0.272 | 10,486,295 | 4,000,107 |
| Buy and hold | 1.15% | 4.90% | 0.234 | 9,318,815 | 3,698,810 |
| Naïve | 0.92% | 4.37% | 0.210 | 2,883,092 | 5,981,745 |
| Dow Jones 30 index (DJ30) | 0.03% | 1.17% | 0.031 | | |

the MV_MS surpasses those of the MV and market indexes. These findings align with past studies that indicate PPP enhances MV performance [3,9,35]. MV_MS also achieves the highest end and average market value. The findings also align with Kim and Suh [22], who suggest that incorporating sentiment analysis enhances the effectiveness of the momentum strategy, particularly in increasing average returns and Sharpe ratios. However, compared to Kim and Suh [22], who use static asset allocation, the MV_MS performs better, providing evidence that sentiment analysis based on the PPP improves over momentum strategy.

In Panel B, using the Dow Jones 30 dataset, MV_MS outperformed MV_M, MV, and the market index, which mirror the outcomes observed with the S&P dataset. Moreover, due to the lower volatility (1.17%) of Dow Jones 30 compared to the S&P 500 Index (1.23%), portfolios utilizing MV_MS, MV_M, and MV exhibit lower standard deviations, though higher average and final market value in the Dow Jones 30 dataset. Except for the final market value of the MV_MS, the market values of all MV-related models in the Dow Jones 30 dataset are higher than their counterparts in the S&P 500 dataset due to large capital composite stock. This finding indicates that MV and MV_M models perform better in more stable markets. In contrast, MV_MS is particularly adept at managing assets in fluctuating markets.

Table 2 Panel A shows that MV_M and MV_MS invest in approximately 50 assets each, integrating the characteristic vector. This finding indicates that MV_MS and MV_M mitigate the issue observed with MV, which tends to invest in only a few assets. Despite MV_MS incurring the highest trade costs and turnover rates, it still performs best (Table 1, Panels A and B) whether using the more stable dataset of the Dow Jones 30 or the more volatile dataset of the S&P 500. The superior performance of the MV_MS stems from its effective asset allocation.

To evaluate the effectiveness of MV_MS during market drawdown, this study examined data spanning the subprime crisis period from January 10, 2007 to December. 30, 2009 (Table 3). Panel A shows that using the S&P 500 dataset, models incorporating the characteristic vector (MV_M, MV_MS) exhibited higher market values and Sharpe ratios compared to MV. Particularly, MV_M outperformed MV_MS slightly. However, when using the Dow Jones 30 dataset, MV_M exhibited a sharp drop in returns, turning negative. MV_MS maintained positive returns and achieved the highest Sharpe ratio.

During the financial crisis (2007–2009), MV_MS maintained positive returns and superior Sharpe ratios compared to the Naïve and buy-and-hold benchmarks. Tables 2 and 4 show that the momentum and momentum-based sentiment strategies (MV_M and MV_MS) require higher trade costs due to adjusting asset allocation more frequently, resulting in higher turnover ratio. These findings confirm that while conventional benchmarks provide simplicity and cost efficiency, the

**Table 2. Comparing trade numbers and cost using the S&P 500 and the Dow Jones 30 datasets (January 1, 2005 to January 4, 2022).**

| | Avg. number of buying assets | Avg. trading cost ($) | Accum. trading costs ($) | Avg. turnover rate (%) |
|---|---|---|---|---|
| Panel A: S&P 500 | | | | |
| MV | 13 | 569 | 118,819 | 0.04% |
| MV_M | 56 | 5,697 | 1,190,600 | 0.10% |
| MV_MS | 48 | 5,911 | 1,235,323 | 0.15% |
| Buy and hold | 382 | 11 | 2,494 | 0% |
| Naïve | 382 | 455 | 100,202 | 0% |
| Panel B: Dow Jones 30 | | | | |
| MV | 8 | 299 | 62,547 | 0.33% |
| MV_M | 17 | 1,065 | 222,666 | 1.47% |
| MV_MS | 14 | 1,172 | 245,013 | 2.39% |
| Buy and hold | 27 | 11 | 2,494 | 0% |
| Naïve | 27 | 309 | 67,923 | 0% |

*Avg. means average. Accum. means accumulated.

proposed MV_MS model delivers more robust and effective portfolio performance, particularly under volatile conditions. This additional validation demonstrates the higher practical value of MV_MS compared to conventional mean-variance models (MV, MV_M), naïve and buy-and-hold strategies.

Similar to Tables 2, 4 indicates that MV tends to concentrate investments in a few assets. Despite MV_MS incurring higher trade costs due to its higher turnover rate, it consistently outperformed other models and the market index. Therefore, MV_MS demonstrates a more stable performance than MV, MV_M, and effectively navigates through the challenges posed by the subprime financial crisis. This finding underscores that integrating the momentum-based sentiment characteristic vector enhances the performance and effectiveness of the mean-variance model in the long term and under drawdown.

**Table 3. Portfolio comparisons using the S&P 500 and the Dow Jones 30 datasets (from Jan. 10, 2007, to Dec. 30, 2009).**

| | Realized | | | Market value | |
| --- | --- | --- | --- | --- | --- |
| | Average Return (%) | Standard deviation (%) | Sharpe ratio | Final | Average |
| Panel A: S&P 500 | | | | | |
| MV | −0.26% | 7.28% | −0.035 | 849,903 | 889,783 |
| MV_M | 0.51% | 9.76% | 0.051 | 996,218 | 961,941 |
| MV_MS | 0.51% | 10.54% | 0.048 | 983,066 | 883,273 |
| Buy and hold | −0.28% | 8.50% | −0.033 | 816,174 | 818,060 |
| Naïve | 0.15% | 0.67% | 0.015 | 899,983 | 835,037 |
| S&P 500 index (S&P500) | −0.01% | 1.89% | −0.006 | | |
| Panel B: Dow Jones 30 | | | | | |
| MV | −0.12% | 9.08% | −0.013 | 842,448 | 843,541 |
| MV_M | −0.37% | 9.37% | −0.039 | 775,966 | 815,582 |
| MV_MS | 0.72% | 10.01% | 0.072 | 1,061,853 | 941,796 |
| Buy and hold | 0.07% | 8.13% | 0.009 | 917,770 | 866,802 |
| Naïve | 0.31% | 8.68% | 0.035 | 970,089 | 879,293 |
| Dow Jones 30 index (DJ30) | 0.01% | 1.73% | −0.004 | | |

**Table 4. Comparing trade numbers and costs using the S&P 500 and the Dow Jones 30 datasets during the financial crisis (from January 10, 2007 to December 30, 2009).**

| | Avg. number of buying assets | Avg. trading cost ($) | Accum. trading costs ($) | Avg. turnover rate (%) |
| --- | --- | --- | --- | --- |
| Panel A: S&P 500 | | | | |
| MV | 11 | 370 | 11,479 | 0.05% |
| MV_M | 51 | 1,668 | 51,727 | 0.11% |
| MV_MS | 36 | 783 | 24,287 | 0.19% |
| Buy and hold | 382 | 80 | 2,494 | 0% |
| Naïve | 382 | 234 | 7,260 | 0% |
| Panel B: Dow Jones 30 | | | | |
| MV | 4 | 809 | 25,075 | 1.54% |
| MV_M | 12 | 205 | 6,361 | 2.08% |
| MV_MS | 13 | 301 | 9,340 | 2.94% |
| Buy and hold | 27 | 80 | 2,494 | 0% |
| Naïve | 27 | 198 | 6,151 | 0% |

We also examine the parameter $\theta$ of the parametric portfolio policy, indicating whether the character vector is used with or without incorporating sentiment information during the subprime crisis period. Using S&P 500 dataset as an example, we present 15 rebalancing results from March 24, 2008, to May 1, 2009, in Table 5. Panel A applies the momentum character vector within a mean-variance (MV) framework, where $\theta$ is positive in seven of 15 rebalancing instances. In contrast, Panel B incorporates sentiment-based momentum, where $\theta$ is equal to 1 in nine out of 15 instances. This finding suggests that the proposed model relies on the sentiment-based momentum character vector more heavily through the parametric portfolio policy, particularly during periods of high market volatility, thereby maximizing investor utility. The influence of

**Table 5. Weights, benchmark weights, and characteristic vector of MV with momentum and sentiment-based momentum.**

**Panel A: MV with momentum allocated on assets of ALB, AVB, CL, and KMX.**

| Date | Weight ($w_i$) | | | | Benchmark weight ($w_i^b$) | | | | | Characteristic vector ($m_i$) | | | |
|---|---|---|---|---|---|---|---|---|---|---|---|---|---|
| | ALB | AVB | CL | KMX | ALB | AVB | CL | KMX | $\theta$ | ALB | AVB | CL | KMX |
| 3/24/08 | 0 | 0 | 0 | 0.011 | 0 | 0 | 0.198 | 0.002 | 0.78 | 0 | 0 | 0 | 0 |
| 4/21/08 | 0 | 0 | 0.010 | 0.016 | 0 | 0 | 0.197 | 0.002 | 0.00 | 0 | 0 | 0 | 0 |
| 5/19/08 | 0 | 0 | 0.010 | 0.019 | 0 | 0 | 0.181 | 0.002 | 0.00 | 0 | 0 | 0 | 0 |
| 6/17/08 | 0 | 0 | 0.010 | 0.018 | 0 | 0 | 0.183 | 0 | 0.00 | 0 | 0 | 0 | 0 |
| 7/16/08 | 0 | 0 | 0.015 | 0.017 | 0 | 0 | 0.176 | 0 | 1.00 | 0 | 0 | 0 | 0 |
| 8/13/08 | 0 | 0 | 0 | 0.020 | 0 | 0 | 0.198 | 0 | 0.00 | 0 | 0 | 0 | 0 |
| 9/11/08 | 0 | 0 | 0 | 0.011 | 0 | 0 | 0.216 | 0 | 1.00 | 0 | 0 | 0 | 0 |
| 10/9/08 | 0 | 0 | 0 | 0.015 | 0 | 0 | 0.222 | 0 | 1.00 | 0 | 0 | 0 | 0 |
| 11/6/08 | 0 | 0 | 0 | 0.017 | 0 | 0 | 0.243 | 0 | 0.00 | 0 | 0 | 0 | 0 |
| 12/5/08 | 0 | 0 | 0 | 0.018 | 0 | 0 | 0.245 | 0 | 1.00 | 0 | 0 | 0 | 0 |
| 1/6/09 | 0 | 0 | 0 | 0.018 | 0 | 0 | 0.241 | 0 | 0.00 | 0 | 0 | 0 | 0 |
| 2/4/09 | 0 | 0 | 0 | 0.020 | 0 | 0 | 0.250 | 0 | 1.00 | 0 | 0 | 0 | 0 |
| 3/5/09 | 0 | 0 | 0 | 0.000 | 0 | 0 | 0.246 | 0 | 1.00 | 0 | 0 | 0 | 0 |
| 4/2/09 | 0 | 0 | 0 | 0 | 0 | 0 | 0.238 | 0 | 0.00 | 0 | 0 | 0 | 0.026 |
| 5/1/09 | 0 | 0 | 0 | 0 | 0 | 0 | 0.221 | 0 | 0.00 | 0 | 0 | 0 | 0.026 |

**Panel B: MV with sentiment-based momentum allocated on assets of ALB, AVB, CL, and KMX.**

| Date | Weight ($w_i$) | | | | Benchmark weight ($w_i^b$) | | | | | Characteristic vector ($m_i$) | | | |
|---|---|---|---|---|---|---|---|---|---|---|---|---|---|
| | ALB | AVB | CL | KMX | ALB | AVB | CL | KMX | $\theta$ | ALB | AVB | CL | KMX |
| 3/24/08 | 0.042 | 0 | 0.030 | 0.010 | 0 | 0 | 0.198 | 0.002 | 1 | 0.083 | 0 | 0 | 0 |
| 4/21/08 | 0.022 | 0 | 0.046 | 0.016 | 0 | 0 | 0.197 | 0.002 | 0 | 0 | 0 | 0 | 0 |
| 5/19/08 | 0 | 0 | 0.111 | 0.019 | 0 | 0 | 0.181 | 0.002 | 0 | 0.083 | 0 | 0 | 0 |
| 6/17/08 | 0.042 | 0.042 | 0.010 | 0.017 | 0 | 0 | 0.183 | 0 | 1 | 0.083 | 0.083 | 0 | 0 |
| 7/16/08 | 0.062 | 0.010 | 0.015 | 0.017 | 0 | 0 | 0.176 | 0 | 1 | 0.083 | 0.083 | 0 | 0 |
| 8/13/08 | 0.030 | 0.015 | 0.107 | 0.020 | 0 | 0 | 0.198 | 0 | 0 | 0 | 0.083 | 0 | 0 |
| 9/11/08 | 0 | 0.049 | 0 | 0 | 0 | 0 | 0.216 | 0 | 1 | 0 | 0.083 | 0 | 0 |
| 10/9/08 | 0 | 0.068 | 0 | 0 | 0 | 0 | 0.222 | 0 | 1 | 0 | 0.083 | 0 | 0 |
| 11/6/08 | 0 | 0.030 | 0.122 | 0 | 0 | 0 | 0.243 | 0 | 0 | 0 | 0 | 0 | 0 |
| 12/5/08 | 0 | 0 | 0.055 | 0.042 | 0 | 0 | 0.245 | 0 | 1 | 0 | 0 | 0 | 0.083 |
| 1/6/09 | 0 | 0 | 0 | 0.061 | 0 | 0 | 0.241 | 0 | 1 | 0 | 0 | 0 | 0.083 |
| 2/4/09 | 0 | 0 | 0 | 0.010 | 0 | 0 | 0.250 | 0 | 1 | 0 | 0 | 0 | 0.083 |
| 3/5/09 | 0.042 | 0 | 0 | 0.017 | 0 | 0 | 0.246 | 0 | 1 | 0.083 | 0 | 0 | 0 |
| 4/2/09 | 0.010 | 0 | 0.119 | 0.019 | 0 | 0 | 0.238 | 0 | 0 | 0 | 0 | 0 | 0 |
| 5/1/09 | 0.015 | 0 | 0 | 0.018 | 0 | 0 | 0.221 | 0 | 0 | 0 | 0 | 0.083 | 0 |

**Table 6. The value of $\theta$ in the S&P 500 and the Dow Jones 30 datasets.**

**Panel A: S&P 500**

| | MV with momentum | | | | MV with sentiment-based momentum | | | |
|---|---|---|---|---|---|---|---|---|
| | $\bar{\theta}$ | $\theta = 0$ | $0 < \theta < 1$ | $\theta = 1$ | $\bar{\theta}$ | $\theta = 0$ | $0 < \theta < 1$ | $\theta = 1$ |
| 1/1/2005-1/4/2022 | 0.35 | 136 | 30 | 54 | 0.72 | 49 | 32 | 139 |
| 1/10/2007-12/30/2009 | 0.31 | 21 | 2 | 8 | 0.65 | 11 | 2 | 18 |

**Panel B: Dow Jones 30**

| | MV with momentum | | | | MV with sentiment-based momentum | | | |
|---|---|---|---|---|---|---|---|---|
| | $\bar{\theta}$ | $\theta = 0$ | $0 < \theta < 1$ | $\theta = 1$ | $\bar{\theta}$ | $\theta = 0$ | $0 < \theta < 1$ | $\theta = 1$ |
| 1/1/2005-1/4/2022 | 0.24 | 153 | 21 | 46 | 0.51 | 90 | 37 | 93 |
| 1/10/2007-12/30/2009 | 0.42 | 17 | 1 | 13 | 0.64 | 11 | 1 | 19 |

sentiment on portfolio rebalancing also results in differing asset allocations, as indicated by the positive weights in Panel B of Table 5 compared to those in Panel A. For example, the allocating weights ($w_i$) of tickers ALB, AVB, CL, and KMX obtained from MV with sentiment-based momentum strategy differ from MV based on the momentum strategy.

Moreover, we compare parameter $\theta$ of the parametric portfolio policy across two rebalancing periods: January 1, 2005, to January 4, 2022 (220 rebalancing instances), and January 10, 2007, to December 30, 2009 (31 rebalancing instances). Table 6 shows that the average value of $\theta$ (denoted as $\bar{\theta}$) is consistently higher under the mean-variance (MV) strategy that incorporates the sentiment-based momentum characteristic vector in both time periods and across the S&P 500 and Dow Jones 30 datasets than under the MV strategy with the traditional momentum vector. The frequency of non-zero $\theta$ values is greater in the sentiment-based momentum strategy. We provide a detailed example of APPLE to illustrate the relationship between $w$, $w^b$, $\hat{m}$ and $\theta$ in S2 Appendix. The findings suggest that integrating sentiment-based momentum into the parametric portfolio policy improves portfolio selection performance.

## 5. Conclusions

This study contributes to the advancement of the parametric portfolio policy by integrating financial news sentiment into the momentum strategy. That is, return momentum is treated as the initial factor in constructing a sentiment-based characteristic vector for asset allocation. Through simulations over a 15-year rebalancing period drawing on S&P 500 and Dow Jones 30 datasets, this study demonstrates that the rebalanced mean-variance model with a momentum-based sentiment vector (MV_MS) enhances returns and market value compared to the momentum-only model (MV_M). It also exhibits resilience in unstable market conditions.

Despite incurring higher trade costs than other mean-variance models, MV_MS achieves the highest Sharpe ratio and market values over the 15-year period. This model is also demonstrated to be more effective during the subprime crisis period: it addresses the issue of concentrated asset investments inherent in conventional mean-variance models by delivering a diversified portfolio with higher Sharpe ratios. Parametric portfolio policy enhances benchmark model performance by incorporating trading strategies and sentiment analysis into the characteristic vector.

For future research, we plan to employ reversal strategies and integrate other unstructured asset attributes, including public sentiment from social media, to optimize characteristic vector selection further. Moreover, alternative benchmark models, e.g., mean-absolute downside risk or conditional value-at-risk, can provide insight into developing higher-performing portfolios.

## Supporting information

**S1 Appendix. Sensitivity analysis of parameter $\alpha$.**
(DOCX)

**S2 Appendix. Relationship among $w$, $w^b$, and $\hat{m}$: the case of Apple.**
(DOCX)

**S1 Data. Data for PLOSone.**
(ZIP)

## Author contributions

**Conceptualization:** Wen-Yi Lee, Yu-Hsuan Lin.

**Data curation:** Wen-Yi Lee, Yu-Hsuan Lin.

**Formal analysis:** Wen-Yi Lee, Yu-Hsuan Lin, Jing-Rung Yu.

**Funding acquisition:** Wen-Yi Lee.

**Investigation:** Yu-Hsuan Lin, Jing-Rung Yu.

**Methodology:** Wen-Yi Lee, Yu-Hsuan Lin, Jing-Rung Yu, Donald Lien.

**Project administration:** Wen-Yi Lee.

**Software:** Wen-Yi Lee, Yu-Hsuan Lin.

**Supervision:** Wen-Yi Lee.

**Validation:** Wen-Yi Lee, Jing-Rung Yu, Donald Lien.

**Writing – original draft:** Wen-Yi Lee, Jing-Rung Yu, Donald Lien.

**Writing – review & editing:** Wen-Yi Lee, Jing-Rung Yu, Donald Lien.

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
