## [Decision Letter · Decision Letter 0]

4 Aug 2025

Dear Dr. Yu,

Thank you for submitting your manuscript to PLOS ONE. After careful consideration, we feel that it has merit but does not fully meet PLOS ONE’s publication criteria as it currently stands. Therefore, we invite you to submit a revised version of the manuscript that addresses the points raised during the review process.

**ACADEMIC EDITOR: **

We look forward to receiving your revised manuscript.

Kind regards,

Jae Wook Song

Academic Editor

PLOS ONE

Journal Requirements: 

[Funding recipient: Wen-Yi Lee

Grant number: NSTC 111-2410-H-141-004-The National Science and Technology Council, Taiwan

URL: https://www.nstc.gov.tw/

The sponsors or funders did not play any role in the study design, data collection and analysis, decision to publish, or preparation of the manuscript.]

 [The authors acknowledge the support provided by the National Science and Technology Council, Taiwan (Grant number: NSTC 111-2410-H-141-004-).]

Reviewers' comments:

Reviewer's Responses to Questions

**Comments to the Author**

1. Is the manuscript technically sound, and do the data support the conclusions?

Reviewer #1: No

Reviewer #2: Partly

2. Has the statistical analysis been performed appropriately and rigorously?

Reviewer #1: No

Reviewer #2: Yes

3. Have the authors made all data underlying the findings in their manuscript fully available?

Reviewer #1: No

Reviewer #2: No

4. Is the manuscript presented in an intelligible fashion and written in standard English?

Reviewer #1: Yes

Reviewer #2: No

Reviewer #1: The paper proposes a method but fails to implement it to some read-world case study. The data used are old, they need to update the references and expand their results. The authors also need to validate their results using some alternative methods in order to show the effectiveness of their results.

Reviewer #2: Comment 1: In the proposed model, the top 30% of assets are first selected based on past returns, followed by an additional filtering of the top 10% with the most positive sentiment. This implies that only the top 3% of assets are finally selected.

On the other hand, in MV_M, the top 10% assets are selected solely based on returns.

This discrepancy in the number of selected stocks makes the comparison between MV_M and MV_MS potentially imbalanced.

It would strengthen the analysis to either (a) tune the 30% and 10% thresholds, or (b) compare both models using the same number of selected stocks.

Comment 2: The parameter alpha, which controls the balance between the current and previous portfolio weights, is fixed at 0.5.

It would be beneficial to include a sensitivity analysis varying alpha to assess its impact on both turnover and portfolio performance.

Comparing turnover rates and returns under different alpha values could add depth to the empirical results.

Comment 3: Please double-check the sign of the last term in Equation (8).

Should it be a plus sign rather than a minus?

Comment 4: The variables l+ and l- appear to represent the quantities of buying and selling, respectively, between consecutive rebalancing periods.

However, this explanation is not clearly articulated in the manuscript.

Adding a concise but clear explanation will help readers better understand the role of these terms in calculating transaction costs.

Comment 5: The description of sentiment score computation is overly brief. Simply stating that NLTK was used is insufficient.

More detail is needed regarding how the sentiment scores were extracted from the text, such as which lexicon or model was applied, how documents were tokenized, and how firm-level sentiment was aggregated.

Comment 6: Please consider including descriptive statistics for the news data used in the sentiment analysis.

This would help validate the robustness and distribution of the sentiment signals.

Comment 7: The manuscript does not clearly explain how each article was mapped to a specific stock.

What criteria were used — such as company name/entity recognition, ticker mentions, or other NLP-based rules?

A brief methodological explanation is needed.

Comment 8: In Table 2, MV_MS holds fewer assets on average than MV_M, yet it incurs higher trading costs and turnover.

This result seems counterintuitive. Could the authors elaborate on why this is the case? A possible explanation might relate to more frequent or aggressive reallocations.

Comment 9: Please check for any typos overall.

"abovementioned", "previou", "Secion", "prposed", "other assets set are as"

**Do you want your identity to be public for this peer review?** For information about this choice, including consent withdrawal, please see our Privacy Policy

Reviewer #1: No

Reviewer #2: No

---

## [Author Response · Author response to Decision Letter 1]

26 Sep 2025

We have attached a completed response letter to reviewers and editor.

---

## [Decision Letter · Decision Letter 1]

13 Oct 2025

Parametric portfolio policy with momentum-based sentiment trading strategy

PONE-D-25-34570R1

Dear Dr. Yu,

We’re pleased to inform you that your manuscript has been judged scientifically suitable for publication and will be formally accepted for publication once it meets all outstanding technical requirements.

Kind regards,

Jae Wook Song

Academic Editor

PLOS ONE

Reviewers' comments:

Reviewer's Responses to Questions

**Comments to the Author**

Reviewer #1: All comments have been addressed

Reviewer #2: All comments have been addressed

2. Is the manuscript technically sound, and do the data support the conclusions?

Reviewer #1: Yes

Reviewer #2: Partly

3. Has the statistical analysis been performed appropriately and rigorously?

Reviewer #1: Yes

Reviewer #2: Yes

4. Have the authors made all data underlying the findings in their manuscript fully available?

Reviewer #1: Yes

Reviewer #2: Yes

5. Is the manuscript presented in an intelligible fashion and written in standard English?

Reviewer #1: Yes

Reviewer #2: Yes

Reviewer #1: The paper has been improved and asnwered to all reviewers' questions. I believe the paper can be published in its present form and would contribute to the field of portfolio optimization.

Reviewer #2: The author has thoroughly addressed all the review comments I provided.

The issues I previously raised regarding the explanation of sentiment data acquisition and processing, as well as the need for additional experiments and sensitivity analyses, appear to have been fully addressed.

**Do you want your identity to be public for this peer review?** For information about this choice, including consent withdrawal, please see our Privacy Policy

Reviewer #1: No

Reviewer #2: No

---

## [Editor Report · Acceptance letter]

PONE-D-25-34570R1

PLOS ONE

Dear Dr. Yu,

I'm pleased to inform you that your manuscript has been deemed suitable for publication in PLOS ONE. Congratulations! Your manuscript is now being handed over to our production team.

Kind regards,

on behalf of

Professor Jae Wook Song

Academic Editor

PLOS ONE